# An Interesting Image of Transmural Migration of a Levonorgestrel-Releasing Intrauterine Device (LNg-IUD)

**DOI:** 10.3390/diagnostics12092227

**Published:** 2022-09-15

**Authors:** Melinda-Ildiko Mitranovici, Diana Maria Chiorean, Adrian-Horațiu Sabău, Iuliu-Gabriel Cocuz, Andreea Cătălina Tinca, Mihaela Cornelia Mărginean, Maria Cătălina Popelea, Traian Irimia, Raluca Moraru, Claudiu Mărginean, Marius Lucian Craina, Izabella Petre, Elena Silvia Bernad, Ion Petre, Ovidiu Simion Cotoi

**Affiliations:** 1Department of Obstetrics and Gynecology, Emergency County Hospital Hunedoara, 14 Victoriei Street, 331057 Hunedoara, Romania; 2Department of Pathology, County Clinical Hospital of Targu Mures, 540072 Targu Mures, Romania; 3Department of Pathophysiology, George Emil Palade University of Medicine, Pharmacy, Science, and Technology of Targu Mures, 38 Gheorghe Marinescu Street, 540142 Targu Mures, Romania; 4School of Medicine, “George Emil Palade” University of Medicine, Pharmacy, Sciences and Technology, 540142 Targu Mures, Romania; 5Faculty of Medicine, “George Emil Palade” University of Medicine, Pharmacy, Sciences and Technology, 540142 Targu Mures, Romania; 6Department of Obstetrics and Gynecology, “George Emil Palade” University of Medicine, Pharmacy, Sciences and Technology, 540142 Targu Mures, Romania; 7Department of Obstetrics and Gynecology, “Victor Babes” University of Medicine and Pharmacy, 2 Eftimie Murgu Sq, 300041 Timisoara, Romania; 8Department of Medical Informatics and Biostatistics, “Victor Babes” University of Medicine and Pharmacy, 2 Eftimie Murgu Sq, 300041 Timisoara, Romania

**Keywords:** intrauterine contraceptive device, transmural migration, embedment, perforation, CT, MRI, ultrasonography

## Abstract

Intrauterine devices (IUDs) are very common as a method of birth control. By adding progesterone (levonorgestrel), a decrease in the risk of complications has been documented, including the risk of perforation. Even though only a few complications have been described, adjacent organs may be involved in the case of migration—a life-threatening situation. A 45-year-old G4P2 woman was seen in our clinic for LNg-IUD removal, according to the medical instructions. Her main complaints were abdominal discomfort, low back pain, and recurrent menorrhagia. A “lost” IUD was initially suspected; the patient confirmed the detection and removal of the control strings, and a subsequent discussion related to delayed transmural migration of the IUD being followed. The ultrasonography revealed the migration of the IUD to the uterine cervix and size-decreased uterine fibroids, confirming the effectiveness of the LNg-IUD. The MRI and ultrasonography being useless, a subsequent X-ray and CT scan were requested, both confirming a myometrium-positioned IUD, adjacent to the serosa and lumbosacral plexus. Even though the IUD is considered a safe device with reversible effect, it can be associated with severe morbidity, with an ultrasound follow-up being required. For more precise detection of the IUD, we strongly recommend an X-ray or CT scan examination, followed by safe removal.

##  

**Figure 1 diagnostics-12-02227-f001:**
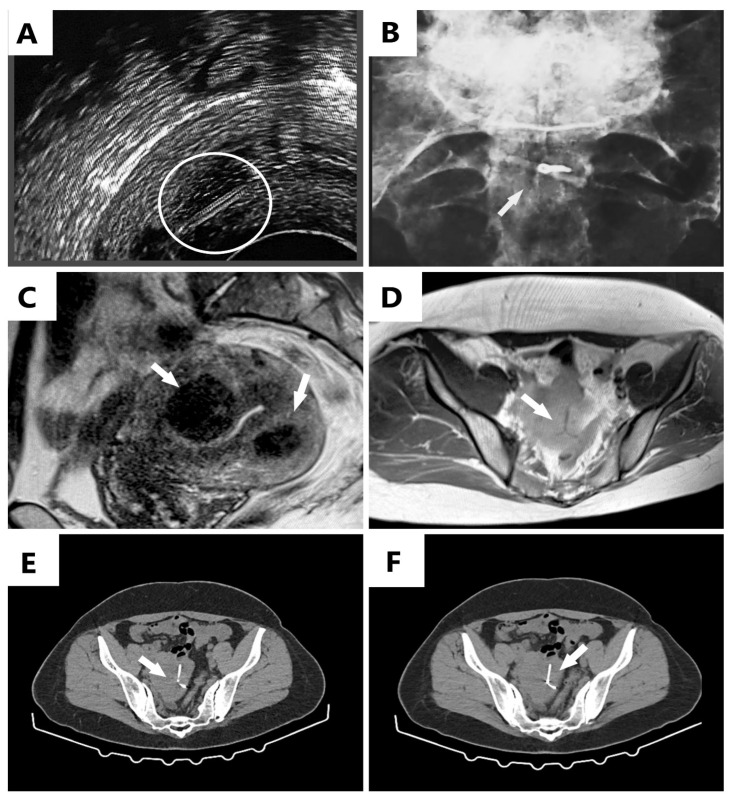
(**A**) Ultrasound examination revealing the control strings of the IUD, initially mistaken for a migrated device marked inside the white circle; (**B**) plain pelvic X-ray revealing the round area of the IUD where the control strings were normally attached (white arrow); (**C**,**D**) magnetic resonance imaging (MRI) scans revealing the uterus with two fibroids (white arrows) and no identified IUD (**C**), and the intramural IUD with a vicious position (upside-down), identified near the serosa (white arrow) (**D**); (**E**,**F**) computed tomography (CT) scan revealing the IUD adjacent to the serosa and lumbosacral plexus (white arrows). A 45-year-old G4P2 woman was seen in our clinic for levonorgestrel (LNg) intrauterine device (IUD) removal according to the medical instructions, having been used for menstrual management. After the insertion of the intrauterine device (IUD), the patient was subsequently followed up with by the gynecology service. Her past medical history included a lumbar discopathy surgery and multiple uterine fibroids, larger than 5 cm in diameter. At the time of admission, the main complaints were about abdominal discomfort, low back pain corresponding to the lumbosacral region of the spine, and recurrent menorrhagia. The ultrasonography revealed what we thought to be the migration of the IUD to the uterine cervix and size-decreased uterine fibroids to 3.5 cm in diameter. The effectiveness of the LNg-IUD was confirmed; not only did it reduce the menorrhagia, but it also reduced the size of the fibroids. An attempt to remove the IUD by extraction failed, obtaining only the control strings of the intrauterine device. Another unsuccessful attempt by hysteroscopy followed, with no evidence of the IUD in the uterine cavity. In the first instance, a “lost” IUD was suspected; however, the patient confirmed the detection and removal of the control strings, a fact that determined a subsequent discussion related to delayed transmural migration of the IUD. Since a subsequent perforation was expected, radiography was requested. Even if the IUD was identified in the myometrium, its vicious position prevented a correct diagnosis: only the part where the strings were attached was visible (Figure 1A), and the device was seen as midline-placed, adjacent to the lumbosacral plex (Figure 1B). Since our patient had in her past medical history a lumbar discopathy surgery, it was considered an explanation for her lumbosacral pains, with no direct causality with the IUD being suspected. A magnetic resonance imaging (MRI) scan by the doctor who performed the hysteroscopy and a computed tomography (CT) scan by the doctor who attempted to remove the IUD were indicated. The MRI scan was declared not very effective at highlighting the device (Figure 1C,D). The CT scan (Figure 1E,F) revealed the myometrium-positioned IUD, adjacent to the serosa and lumbosacral plexus, a fact that explained the patient’s lumbar pain. Furthermore, subsequent removal by laparoscopic hysterectomy with an eventual conversion to laparotomy in the case of failure was proposed. This maneuver was successfully performed, and the patient easily recovered without difficulties, the back pain has subsided. To our knowledge, complications occur rarely, of 1/1000 insertions [1], but these can be severe and life-threatening, such as intramural IUD migration, with one of two manifestations: embedment, in which case it does not exceed the serosa, remaining at the level of myometrium, or perforation, in which case it can migrate to the organs inside the peritoneal cavity, a situation encountered in 85% of cases [1]. Another complication described was related to an intrauterine device identified within an ovarian tumor, in a 63-year-old patient, without a clear explanation as to whether the tumor was a consequence of the migration of the IUD inside the ovary or its development was concomitant [1]. The mechanism of migration remains unknown [1], but uterine contractions are thought to play a role [2,3,4]. According to Chai et al., two devices encountered in the same patient were described as migrating devices, the second one being inserted during a cesarean section procedure, both exceeding the uterine serosa and discovered on CT scan, the second one being inserted after considering the first one, erroneously, “lost” [5]; this incident draws attention to the importance of an accurate diagnosis before making a decision to mount another intrauterine device. As in our patient’s case, migration of the IUD can occur over time as a distant migration, a result of gradual erosion of the myometrium [2,3], or it can occur immediately after insertion, this phenomenon being encountered in the immediate postpartum period. According to World Health Organization (WHO) recommendations, the indicated treatment is extraction of the IUD, as soon as possible; if we are not able to retrieve the device, a laparoscopy should be considered [6,7,8,9,10]. Laparotomy should remain the optimal option for most of the complicated cases [3]. In case of migration in areas such as the rectosigmoid, the colon, or the urinary bladder, other minimally invasive methods should be considered, such as colonoscopy and cystoscopy, as more appropriate [11,12,13,14,15]. The strong point of our case report is represented by the levonorgestrel-releasing IUD detected as migrated before perforating the serosa and bringing about serious consequences for the patient. As levonorgestrel-releasing IUDs are considered useful for the management of menometrorrhagia, these patients require careful follow-up, performing an ultrasound examination of the IUD in the uterine cavity. We strongly recommend that when the IUD cannot be visualized by ultrasound and the control string cannot be seen, it should not be considered as having been expelled. We suggest a subsequent radiographic examination or a CT scan for adequate detection of its localization, followed by safe removal.

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
