# Peer review of "An Interesting Image of Transmural Migration of a Levonorgestrel-Releasing Intrauterine Device (LNg-IUD)"

_diagnostics, 2022, doi:10.3390/diagnostics12092227_

Round 1
Reviewer 1 Report
The manuscript presents a very interesting clinical case and is, in general, well written with occasional typographical or grammatical errors. For instance, in Line 56, it should be "lumbar" instead of "lombar". In Line 77, "histerectomy" should be replaced with "hysterectomy". The images are well presented and discussed.
Author Response
Thank you very much for your review! I requested editing in English!

Reviewer 2 Report
1. There are some spelling errors:
56th line: …. Her past medical history included a “lumbar “
77th line: … removed by laparoscopic “hysterectomy “
2. IUD displacement is not a rare situation in clinical scenario. There are already some flow charts established for management of a translocated IUD. What is the reason you choose MRI rather than CT scan when you have the first instance of a missed IUD? Besides, usually, MRI is better than CT scan for evaluating soft tissue lesion. Why did you find the IUD was myometrium-positioned and adjacent to “lumbosacral plexus” after CT scan? In your figure E and F, there is no obvious relationship between this IUD and lumbosacral plexus.
3. If the IUD already penetrated to serosal layer of uterus, why did you need to perform a laparoscopic hysterectomy in order to remove IUD?
4. Did her symptoms of lower back pain relieved after hysterectomy? And did she really have a lumbar discopathy? If her lower back pain was not related to lumbar discopathy, why did she still receive lumbar surgery?
5. The percentage or incident of migration of IUD and further management can also be mentioned inside the discussion
Author Response
Cover letter:
1). I requested an English revision.
2). As a rarity, migration occurs in 1/1000 insertions of IUDs, of these 85% perforate organs [1]. Unfortunately, the established diagnosis in these situations, especially if the control strings are not visible, is expulsion. The diagnostic algorithm was the following: ultrasound ( line 59), we saw the string and we misinterpreted it as the IUD, followed by hysteroscopy (line 63) with no evidence of IUD in uterine cavity, then X ray (line 66) to find IUD and the device was seen as I described in the text and we can see in the Figure 1B, then we requested a MRI (line 72) to find its exact location in the myometrium, the MRI image was not satisfactory, as we can see on the image, reason why I asked for a CT scan (line77). The IUD was not found in the uterine cavity by hysteroscopy, so we tried to show clear enough the missed IUD, and as we can see on the images, the CT scan is more efficient. In the figures C and E we can see the relationship between the uterus and the lumbosacral plexus (C), and the short arm of the device near the uterine serosa under the fibroid, suggesting a relationship with the lumbosacral plexus. In the clinical context where the patient complains of back pain that subsides after surgery, we demonstrate that the relationship is real.
3). The IUD did not penetrate the serosal layer of uterus. There are two reasons for hysterectomy: the first is the recurrent menorrhagia with the fibroids, the second is according to World Health Organization (WHO) recommendations, the indicated treatment is the extraction of the IUD, as soon as possible; if we are not able to retrieve the device, a laparoscopy should be considered [6,7,8,9,10]. (line 94).
4). The back pain has subsided after hysterectomy, she had in her past history a lumbar discopathy surgery. At first I thought that the pain was due to recurent discopathy, but the clinical context and the imaging aspects leads to the conclusion that the device was really close to lumbosacral plexus.
5).The incident of migration is of 1/1000 insertions, sometimes remaining at the level of myometrium, or perforation – in which case it can migrate to the organs inside the peritoneal cavity, situation encountered in 85% of the migration cases [1] (line 79,81-83).I added the details in the manuscript.
Thank you for reviewing my manuscript. I hope I could explain everything here.

Reviewer 3 Report
I found the case report interesting and suitable to be considered for publication. However, I think there are significant issues to be fixed before further considering it for publication.
1) The article structure should follow the CARE indication described in the appropriate checklist: Gagnier JJ, Kienle G, Altman DG, Moher D, Sox H, Riley D; CARE Group. The CARE guidelines: consensus-based clinical case reporting guideline development. BMJ Case Rep. 2013 Oct 23;2013:bcr2013201554. doi: 10.1136/bcr-2013-201554. PMID: 24155002; PMCID: PMC3822203.
2) A timeline Table or plot could be an interesting improvement.
3) According to your case report, you can conclude about morbidity but not mortality. Please milden the conclusion [more general issues can be discussed within the discussion section].
4) The Figure 1 caption and case presentation are pasted in the same paragraph. Please fix the issue and follow the CARE guidelines for the sections to be included in a case report.
Author Response
Cover letter:
1). The article structure is according to the requested topic , which is” Interesting images”, as I wrote in the upper left corner of the manuscript. I am quoting the editor who asked me for this type of article, after I sent the requested abstract:
< An example of an interesting images paper: https://www.mdpi.com/2075-4418/11/4/672/htm, Diagnostics encourages the submission of Interesting Images. The number of images is at the discretion of the author. No regular manuscript text (introduction/methods/results/discussion) should be included. Instead, images should be accompanied by detailed legends with no restriction in length. Reference citations should appear in the legends.>
2). I tried to emphasize the chronology in the Abstract, so as to respect the templates requested for this type of article (line 31-38, 40-42). These are detailed in the legend. The templates do not allow what you ask.
3). I exclude mortality from the text as you suggested. I started with a “ case report “ type article, then , at the request of the editor, I transformed it into “ Interesting images “ type manuscript, with no Discussions or Conclusions in the text. I even emphasized a conclusion in the Abstract and we also excluded 15 references .
4). In this type of manuscript all the images are put together, followed by a detailed legend, as the requirement is presented in the quote, seen above in paragraph 1. < >
Thank you very much for reviewing the manuscript. I hope I could explain everything here.
Round 2
Reviewer 3 Report
The manuscript is significantly improved and suitable to be published.